# Based on HPLC and HS-GC-IMS Techniques, the Changes in the Internal Chemical Components of *Schisandra chinensis* (Turcz.) Baill. Fruit at Different Harvesting Periods Were Analyzed

**DOI:** 10.3390/molecules29081893

**Published:** 2024-04-22

**Authors:** Bowei Sun, Yiping Yan, Mingjie Ma, Jinli Wen, Yanli He, Yining Sun, Pengqiang Yuan, Peilei Xu, Yiming Yang, Zihao Zhao, Li Cao, Wenpeng Lu

**Affiliations:** 1Institute of Special Animal and Plant Sciences, Chinese Academy of Agricultural Sciences, Changchun 130112, China; 2022050841@ybu.edu.cn (B.S.); 82101225211@caas.cn (Y.Y.);; 2College of Agriculture, Yanbian University, Yanji 133002, China; 3Faculty of Agriculture, Jilin Agricultural Science and Technology University, Jilin 132101, China

**Keywords:** *Schisandra chinensis*, harvest time, lignans, organic acids, volatile compounds

## Abstract

*Schisandra chinensis*, as a traditional Chinese herbal medicine, has clear pharmacological effects such as treating asthma, protecting nerves and blood vessels, and having anti-inflammatory properties. Although the *Schisandra chinensis* fruit contain multiple active components, the lignans have been widely studied as the primary pharmacologically active compound. The volatile chemical components of *Schisandra chinensis* include a large amount of terpenes, which have been proven to have broad pharmacological activities. However, when to harvest to ensure the highest accumulation of pharmacologically active components in *Schisandra chinensis* fruits is a critical issue. The *Schisandra chinensis* fruit trees in the resource nursery were all planted in 2019 and began bearing fruit in 2021. Their nutritional status and tree vigor remain consistently good. The content of lignans and organic acids in the fruits of *Schisandra chinensis* over seven different harvest periods was tested, and the results of high-performance liquid chromatography (HPLC) indicated that the lignan content was higher, at 35 mg/g, in late July, and the organic acid content was higher, at 72.34 mg/g, in early September. If lignans and organic acids are to be selected as raw materials for pharmacological development, the harvest can be carried out at this stage. Using HS-GC-IMS technology, a total of 67 volatile flavor substances were detected, and the fingerprint of the volatile flavor substances in the different picking periods was established. It was shown by the results that the content of volatile flavor substances was the highest in early August, and 16 flavor substances were selected by odor activity value (OAV). The variable importance in projection (VIP) values of 16 substances were further screened, and terpinolene was identified as the key volatile flavor substance that caused the aroma characteristics of *Schisandra chinensis* fruit at different harvesting periods. If the aroma component content of *Schisandra chinensis* fruit is planned to be used as raw material for development and utilization, then early August, when the aroma component content is higher, should be chosen as the time for harvest. This study provides a theoretical basis for the suitable harvesting time of *Schisandra chinensis* for different uses, and promotes the high-quality development of the *Schisandra chinensis* industry.

## 1. Introduction

*Schisandra chinensis* (Turcz.) Baill., a perennial woody vine indigenous to China and prevalent throughout Southeast Asia, is esteemed in traditional Chinese medicine. The dried mature fruits, known as ‘Northern *Schisandra chinensis*’, are utilized medicinally and acclaimed for their therapeutic efficacy and minimal toxicity in treating various ailments, thereby possessing significant medicinal value [1]. The fruit, rich in active constituents like lignans, organic acids, polysaccharides, and triterpenoids, is traditionally employed for liver protection, immune enhancement, and cardiovascular defense [2]. With its distinctive aroma and notable medicinal properties, *Schisandra chinensis* fruit is garnering growing interest for its health and wellness benefits, highlighting its considerable research importance in medical and healthcare product development, as societal emphasis on health intensifies. Schisandrin, a principal component, has been demonstrated to modulate intestinal flora, consequently diminishing hippocampal neuron loss and ameliorating the cognitive deficits linked to Alzheimer’s disease [3]. Additionally, schisandrin A is recognized for attenuating ferroptosis and necroptosis triggered by hyperglycemia, conferring anti-inflammatory advantages and presenting a potential diabetes therapy [4]. Moreover, schisandrin A has been reported to mitigate asthma symptoms through its anti-inflammatory action [5]. Schisantherin A markedly curtails oxidative stress and inflammation induced by isoproterenol, offering prophylactic protection against myocardial infarction-induced cardiac injury [6]. Schisandrin B, within the body, suppresses cellular proliferation and prompts apoptosis by influencing signaling cascades, suggesting a viable therapeutic strategy for colon cancer tumor suppression [7]. Thus, lignans are the foremost pharmacologically active constituents of *Schisandra chinensis*, and this investigation focuses on alterations in the lignans and associated volatile compounds, excluding other phenolic substances.

At present, the predominant analytical techniques for lignans and organic acids in fruits encompass gas chromatography (GC) [8], capillary electrophoresis (CE) [9], ion chromatography (IC) [10], and high-performance liquid chromatography (HPLC) [11]. While GC is highly sensitive, it necessitates extensive sample preparation and may yield skewed results for high molecular weight organic acids. CE is economical but lacks consistency, and its quantitative precision is suboptimal. IC is accurate yet time-consuming, rendering it impractical for swift sample analysis. HPLC, employing a liquid mobile phase, is versatile and reliable, capable of the concurrent detection of diverse lignans and organic acids, thus becoming a prevalent analytical method in disciplines like agronomy, chemistry, and medicine [12].

In recent years, the primary methods for detecting volatile substances in products such as fruits, foods, and medicinal materials include gas chromatography–olfactory mass spectrometry (GC-O-MS), gas chromatography–mass spectrometry (GC-MS), and gas chromatography–ion mobility spectrometry (GC-IMS). GC-MS is widely used due to its established technology and extensive spectral library, but it struggles with complex sample pretreatment and isomer resolution [13]. GC-IMS, an emerging visualization detection technology, is lauded for effectively identifying volatile components in fruits [14,15,16]. It offers high sensitivity, selectivity, resolution, and rapid analysis speed, requires no sample concentration or enrichment, reduces sample pretreatment time, maintains flavor substance stability, and is commonly used for trace volatile organic compound detection across various research fields. GC-IMS is often paired with headspace solid-phase microextraction (HS-SPME) technology, combining GC’s separation capabilities with IMS’s sensitivity and speed to detect low-concentration volatile components that contribute significantly to flavor. This combination enhances qualitative analysis precision, and the results can benchmark quality assessment. Its application is extensive in analyzing and identifying volatile components in fruits, Chinese medicinal materials, and food.

Timely harvesting is crucial for fruit quality. Currently, the harvesters judge ripeness based on fruit appearance, such as size and color, which does not accurately reflect nutritional quality. Early harvesting prevents full development and sufficient internal dry matter accumulation, failing to highlight the fruit’s inherent qualities. Conversely, late harvesting misses the peak nutrient accumulation period. Thus, identifying the optimal harvesting time is vital to ensure that the unique flavor quality and pharmacological activity of the fruits are retained. Recent research on *Schisandra chinensis* has focused on its pharmacological components, with few studies on the harvesting period. Currently, most of the *Schisandra chinensis* used in China are live seedlings, and there is an urgent need to promote high-quality varieties. This study uses ‘Yanzhihong’, a new variety approved by the Institute of Special Products of the Chinese Academy of Agricultural Sciences, as the test material. ‘Yanzhihong’, a superior *Schisandra chinensis* variety, exhibits a higher lignan content than the pharmacopeial standard and demonstrates better stress resistance and yield compared to other varieties [17,18]. The study aims to determine its lignans, organic acids, and volatile flavor substances using HPLC and GC-IMS technology, providing a theoretical basis for the optimal harvest time of *Schisandra chinensis* for various uses and promoting the high-quality development of the industry.

## 2. Results and Discussion

### 2.1. Analysis of Lignans in Fruit of Schisandra chinensis at Different Harvesting Stages

*Schisandra chinensis* contains a significant amount of pharmacologically active components, with the majority being phenolic substances such as flavonoids, lignans, and anthocyanins. Lignan is the main pharmacologically active ingredient in *Schisandra chinensis;* it has shown a good curative effect on many diseases [3,4,5,6,7]. Therefore, this experiment conducted an in-depth study on lignans. The content of schisandrin should not be less than 0.40% in the *Pharmacopoeia of the People’s Republic of China* (2020 edition). The changes in lignan content in *Schisandra chinensis* in different harvesting periods are shown in Table 1. The six lignans showed a decreasing trend in different degrees during the whole growth period. The highest content of schisandrin was 21.70 mg/g on 25 July, and the content of schisandrin continuously decreased as the harvesting period went on, reaching the lowest point of 10.25 mg/g on 6 September, and the subsequent content began to rise, and the content of schisandrin was 10.71 mg/g on 13 September. The decreasing trend in schisandrol B content was the same as that of schisandrin; the highest content was 1.34 mg/g on 25 July, the lowest content was 0.64 mg/g on 6 September, and the content rose to 0.71 mg/g on 13 September. After falling to a low on 22 August, schisantherin A began to rise, reaching 1.83 mg/g on 13 September. The change trend in Schisantherin B was consistent with that of ester schisandrin and schisandrol B. 25 July was the highest period, and the content began to rise after the content dropped to the lowest on 6 September. The change trend in schisandrin A and schisandrin B content was the same as that of schisantherin A, the content of lignan dropped to the lowest point on 22 August, and then increased in the subsequent harvesting period. The main reason for the decrease in lignan content is that the lignan mainly exists in the seeds. With the increase in fruit maturity, the proportion of pulp increases, resulting in a trend of a decrease in lignan content with the later harvest period. At the same time, lignan components participate in the synthesis of other medicinal components at the maturity stage, resulting in a gradual decrease in the content.

### 2.2. Analysis of Organic Acids in Fruit of Schisandra chinensis at Different Harvesting Stages

Organic acids, which are organic compounds with acidic properties, were found to be widely present in nature. Natural organic acids that possess certain physiological activities, such as citric acid and malic acid, were distributed extensively in Chinese herbs and fruits, providing pharmacological effects including expectorant, antitussive, and vasodilating properties [19]. As indicated in Table 2, the higher concentration of organic acids in *Schisandra chinensis* contributed to the fruit’s sour taste. The quinic acid content in *Schisandra chinensis* was initially low in July, increased to 0.52 mg/g on 1 August, then began a continuous decline, reaching 0.42 mg/g on 22 August, with a brief increase noted on 6 September. The L-tartaric acid content was measured at 1.33 mg/g on 25 July, fell to its lowest of 1.27 mg/g on 1 August, and fluctuated throughout the subsequent harvesting period, peaking at 1.73 mg/g on 6 September. L-malic acid, the most significant organic acid in *Schisandra chinensis*, had a content of 29.20 mg/g on 25 July. It also began to decline during the growth and development process, reaching the lowest point of 26.50 mg/g on 9 August. During the later maturity period, the content gradually accumulated, with the highest recorded at 36.20 mg/g on 6 September. Citric acid exhibited an increasing trend throughout the harvest period, starting from the lowest content of 16.92 mg/g on 25 July and rising to 33.93 mg/g on 6 September.

Organic acids are found to exist widely in nature, and the differences in the composition and content of organic acids play a crucial role in determining the sweet and sour flavors of fruits. The interaction with sugars gives fruits unique flavor characteristics. To further study the relationship between the four organic acids, a correlation analysis of the obtained data was conducted. As demonstrated in Table 3, citric acid was positively correlated with quinic acid, tartaric acid, and malic acid, indicating that citric acid plays a leading role in the growth and development of *Schisandra chinensis* fruit. Based on the accumulation of organic acids during the fruit ripening process, fruits could be categorized into malic acid type, citric acid type, and tartaric acid type. The results showed that *Schisandra chinensis* belonged to the citric acid type of fruit.

### 2.3. Analysis of Volatile Flavor Substances in Fruit of Schisandra chinensis in Different Harvesting Periods

#### 2.3.1. Fingerprint of Volatile Substances in Fruit of *Schisandra chinensis* in Different Harvesting Periods

The fruit of *Schisandra chinensis* possesses a unique aroma, and the variation in the types and contents of volatile flavor substances within the fruit have a certain impact on its quality. To analyze the differences in volatile flavor substances in the fruit of *Schisandra chinensis* across various harvesting periods, samples from seven distinct harvesting times were examined, and each group’s samples were measured thrice in parallel to establish the fingerprint of volatile flavor substances for different harvesting periods. The intensity of the color correlated with the strength of the signal peak and the content level. The composition and variation of volatile flavor substances in *Schisandra chinensis* at different harvesting periods (Appendix A) were discerned through fingerprinting. As illustrated in Figure 1, with the increase in fruit ripeness, the content of Methyl 2-furoate, 2-Octanone, (Z)-3-Hexenyl propionate, 2-butanol, Cyclooctanol, 4-Hydroxy-4-methyl-2-pentanone, and other substances increased. Conversely, 3-Ethyl-2-hydroxy-2-cyclopenten-1-one, 6-Methyl-5-hepten-2-one, (2E,6Z)-Nona-2,6-dien-1-ol, Beta-ocimene, (E)-Ethyl-2-hexenoate, 2-Undecanone, and other substances exhibited a decreasing trend.

#### 2.3.2. Two-Dimensional Spectra of Volatile Substances of Fructus *Schisandra chinensis* in Different Harvesting Periods

The two-dimensional spectra of volatile flavor substances in *Schisandra chinensis* fruit are displayed in Figure 2. The red vertical line at the horizontal coordinate 0.5 represents the reactive ion peak (RIP peak), where the horizontal coordinate indicates the ion migration time, and the vertical coordinate denotes the gas chromatographic retention time. The points on both sides of the RIP peak are the detected volatile flavor substances. Differences were noted in the fingerprint spectra of the seven *Schisandra chinensis* during the picking period, primarily in the content. The color signifies the concentration of detected substances: white indicates lower content, red suggests higher content, and the darker the color, the greater the substance content.

The difference spectra of the volatile flavor substances in *Schisandra chinensis* fruit are depicted in Figure 3. Using the fruit of *Schisandra chinensis* from 25 July as a reference, the difference spectra were derived by subtracting the signal peaks from that date. The blue area indicates that the content of the substance during this period was lower than on 25 July, while the red area signifies that the content was higher than on that date, with the intensity of the color reflecting the magnitude of the difference. The difference spectra revealed that on 25 July, the contents of alpha-Pinene, beta-Ocimene, Ethyl 3-hydroxybutyrate, and Isoamyl propionate in *Schisandra chinensis* fruit, along with Cis-6-Nonen-1-ol, tert-Butanol, 3-Methyl-3-buten-1-ol, 2-Butanone, and 3-Ethyl-2-hydroxy-2-cyclopenten-1-one were higher compared to other stages of *Schisandra chinensis*.

#### 2.3.3. Volatile Flavor Components

Using C4–C9 ketone as an external standard method, the GC retention time and IMS migration time for each volatile flavor substance were examined by VOCal. Quantitative and qualitative analyses of volatile flavor substances in *Schisandra chinensis* fruit were performed using the NIST and IMS databases integrated into GC-IMS. As illustrated in Figure 4, a total of 67 volatile flavor substances were detected; among these, 17 were terpenes, followed by 15 ketones, 14 esters, 10 alcohols, 5 aldehydes, and 3 acids, with one each detected for pyrazines, quinolines, and furans. These substances interact to form the unique flavor of the *Schisandra chinensis* fruit. Terpenes accounted for 40.64–42.95% of the total volatile flavor substances, with esters comprising 29.97–32.55%. Terpenes and esters were identified as the main aroma substances in *Schisandra chinensis* fruit.

Terpene substances make up the largest proportion in *Schisandra chinensis* fruit, and their aroma is mainly characterized by fruity scents, including those of citrus, apple, and lemon. Among the terpenes detected, Alpha-farnesene and terpinolene contributed citrusy odors, laurene had a spice odor, beta-Ocimene had an apple odor, Beta-pinene came with a woody smell, and G-terpinene and limonene bore a lemon smell. During the seven harvesting periods, the highest terpene flavor substance content was recorded at 59,167.52 μg/kg on 22 August, followed by 59,081.87 μg/kg on 1 August, 58,333.39 μg/kg on 9 August, 57,866.47 μg/kg on 6 September, 57,795.17 μg/kg on 14 August, 57,275.49 μg/kg on 25 July, and 56,331.19 μg/kg on 13 September.

The aroma of the esters was mainly fruity and grassy. (E)-Ethyl-2-hexenoate and (Z)-3-Hexenyl propionate, known for their green grass aroma, were detected in the fruits of *Schisandra chinensis*. Alpha-Angelica lactone, which carries a medicinal aroma, was also identified in *Schisandra chinensis* fruit. The floral-scented dihydrocarveol acetate, the fruit-scented isobutyl propanoate and isoamyl propionate, along with the pleasure-scented ethyl formate and methyl 2-furoate, were part of the aromatic profile. On 13 September, the highest content of ester flavor substances was recorded at 45,108.55 μg/kg. This was followed by 44,875.9 μg/kg on 1 August, 43,820.12 μg/kg on 6 September, 43,019.52 μg/kg on 25 July, 42,223.01 μg/kg on 9 August, 40,336.78 μg/kg on 14 August, and 40,019.15 μg/kg on 22 August.

Alcohol substances accounted for 8.20–9.58% of the total volatile flavor substances of *Schisandra chinensis* fruit, and its aroma was mainly reflected in the spicy and fruit aromas. 2-butanol with a wine odor, tert-Butanol with an alcohol odor, and cineole with a spicy and cool taste were detected in *Schisandra chinensis* fruit. Cis-6-Nonen-1-ol with a fruity aroma and (2E,6Z)-Nona-2, 6-dien-1-ol with a grassy aroma. The date with the highest alcohol flavor substance content was 25 July with 13,017 μg/kg. It was followed by 12,555.82 μg/kg on 1 August, 12,358.55 μg/kg on 9 August, 12,227.15 μg/kg on 22 August, 11,664.35 μg/kg on 14 August, 11,542.67 μg/kg on 13 September, and 11,246.33 μg/kg on 6 September.

A total of 15 ketones were detected, with a large number of species but a relatively small proportion, 7.09% to 8.45%, and their aroma is mainly embodied as fruit, such as 6-Methyl-5-hepten-2-one, 2-Nonanone, 2-Octanone, 2-Undecanone, etc. The highest ketone flavor substance content was 11,955.96 μg/kg on 1 August. It was followed by 10,968.38 μg/kg on 9 August, 10,596.004 μg/kg on 13 September, 10,460.02 μg/kg on 14 August, 10,371.49 μg/kg on 22 August, 9830.23 μg/kg on 25 July, and 9719.44 μg/kg on 6 September.

The contents of aldehydes, acids, pyrazines, quinolines and furans were relatively small, accounting for 5.47–6.79%, 0.45–0.51%, 0.52–0.58%, 3.12–3.44%, 0.05–0.12% of the total volatile flavor substances.

#### 2.3.4. Principal Component Analysis (PCA) of *Schisandra chinensis* Fruit

Principal component analysis is a widely used multivariate statistical analysis method. All variables identified in the sample were formed into a new set of comprehensive variables, from which 2–3 comprehensive variables were extracted to reflect the information of the original variables. The differences between the samples were assessed according to the contribution rate of the extracted comprehensive variables, visualizing the redundant data. The distance of the different groups of samples reflected the differences in the formation of the communities [20]. As shown in Figure 5, with 67 flavor substances detected as dependent variables and 7 harvesting dates as independent variables, multivariate statistical analysis was conducted on the volatile flavor substances of *Schisandra chinensis* fruit at different harvesting dates. The seven harvesting periods showed an obvious separation trend on the two-dimensional map, and a total of two principal components were extracted, with the contribution rate of PC1 being 45.9% and that of PC2 being 22%. PCA could directly reflect the difference in the flavor substance content in the *Schisandra chinensis* fruit between the different harvesting periods. As illustrated in the figure, the distance between fruits on 25 July and 1 August was smaller, and the distance between fruits on 13 September and other harvesting periods was greater, indicating that there were significant differences in aroma characteristics between the different harvesting periods.

The data obtained from the principal component analysis underwent further processing and analysis. As depicted in Figure 6, in the orthogonal partial least squares discriminant analysis (OPLS-DA), the fitting index of the independent variable R^2^X was 0.9, the fitting index of the dependent variable R^2^Y was 0.817, and the model prediction index Q^2^ was 0.545; all were greater than 0.5, making the fitting results acceptable. To further confirm the model’s validity, 200 permutation analyses were conducted on the data. The results, presented in Figure 7, show that the intersection point of the regression line of Q^2^ and the vertical axis was less than 0, indicating that the model was effective, with no overfitting, and that the analysis was reliable and applicable.

#### 2.3.5. Fruit Odor Activity Value (OAV) Analysis of *Schisandra chinensis*

The substances detected by GC-IMS were analyzed by one-way ANOVA. The flavor substance with a *p*-value less than 0.05 was screened, and the corresponding threshold value of the flavor substance in air was determined according to Odor Threshold: Compilation of Odor Thresholds in Air, Water and Other Media (2011 edition), and its OAV value was calculated. The level of flavor substance content could not be directly used as the basis of fruit aroma characteristics but depended on its concentration and odor threshold. It is generally believed that flavor substances with an OAV value greater than 1 contribute more to fruit aroma and play a key role in the aroma system [21]. As shown in Table 4, a total of 16 flavor substances with OAV values greater than 1 were detected in the seven harvesting periods, after calculation. Among them, five terpenes were the most abundant, including alpha-Pinene, terpinolene, myrcene, G-terpinene, and limonene. The three esters were (E)-ethyl-2-hexenoate, bornyl acetate, and isobutyl propanoate. The alcohols included (2E,6Z)-Nona-2,6-dien-1-ol, cineole, and 2-butanol. The four ketones were 6-methyl-5-hepten-2-one, 2-nonanone, mesityl oxide, and 2-octanone; and one aldehyde was citronellal. Overall, the OAV value of alcohols was significantly higher than that of other flavor substances, with the maximum OAV value of (2E,6Z)-Nona-2,6-dien-1-olranging between 468.78–2682.15, which contributed more to the overall aroma, followed by cineole with an OAV of 495.72–729.44. Combining multiple databases to describe the two substances presented a grassy aroma and a spicy cool taste. The spicy cool flavor was an important aroma characteristic of *Schisandra chinensis*.

The value of VIP (variable important in projection) was the weight value of OPLS-DA model variables, which reflected the influence intensity of each variable on the sample. It was generally based on the VIP value of >1. The data of flavor compounds with a *p*-value < 0.05 and OAV value > 1 were further quantified in *Schisandra chinensis* fruit at seven stages, and OPLS-DA analysis was performed as a variable to accurately screen out the VIP value > 1 substance as the key volatile flavor substance of *Schisandra chinensis.* The results are shown in Figure 8. A total of six volatile flavor substances that contributed more to *Schisandra chinensis* were screened: terpinolene, citronellal, 2-butanol, (E)-ethyl-2-hexenoate, (2E,6Z)-Nona-2,6-dien-1-ol, and 2-octanone, which may be markers of the distinctive flavor of *Schisandra chinensis* fruit. Among them, terpinolene with the highest VIP value was one of the key markers that caused the aroma characteristics of *Schisandra chinensis* fruit in different harvesting periods.

## 3. Materials and Methods

### 3.1. Experimental Materials

‘Yanzhihong’, a new variety approved by the Specialty Research Institute of the Chinese Academy of Agricultural Sciences, was obtained from the National Forest Germplasm Resource Bank of *Schisandra chinensis* (Figure 9). *Schisandra chinensis* was planted in 2019, with a total of 6 trees of consistent age. The harvest period was from July to September 2022, and the fruit was picked once every seven days from the color transformation stage to the fruit maturity stage. Seven periods were selected: 07–25, 08–01, 08–09, 08–14, 08–22, 09–06, 09–13. Fruit trees that showed good growth in the resource nursery were randomly selected, and 50 *Schisandra chinensis* fruits with uniform size and free from pests and diseases were also chosen. After being dried to constant weight, they were used to determine the indexes of *Schisandra chinensis*.

### 3.2. Reagents and Instruments

Test reagents: schisandrin standard, schisandrol B standard, schisantherin A standard, schisantherin B standard, schisandrin A standard, schisandrin B standard, quinic acid standard, citric acid standard, L-tartaric acid standard, L-malic acid standard were chromatopure.

Instruments: 1/10,000 balance (Model BSA 224S-CW, Sartorius Scientific Instruments Ltd., Beijing, China.; ultrasonic cleaner (Model KQ-300E, Kunshan Ultrasonic Instrument Co., Ltd., Kunshan, China); electric thermostatic water bath (Model HWS24, Shanghai Yiheng Scientific Instrument Co., Ltd., Shanghai, China); high-speed refrigerated centrifuge (Model Allegra 64R, Beckman Coulter, Inc., Carlsbad, CA, USA); high-speed multi-functional grinder (Model TQ-500Y, Yongkang Tianqi Shengshi Industry and Trade Co., Ltd., Yongkang, China); high-performance liquid chromatograph (Model G1329A, Agilent Technologies Co., Ltd., Waldbronn, Germany); FlavourSpec^®^ Flavor Analyzer (G.A.S Co., Ltd., Jinan, China).

### 3.3. Experimental Methods

#### 3.3.1. Lignan Content Detection

Preparation of reference solution: precision weighing the control products schisandrin 1.00 mg, schisandrol B 1.43 mg, schisantherin A 1.24 mg, schisantherin B 1.44 mg, schisandrin A 1.25 mg, schisandrin B 1.29 mg, and adding met7hanol into a 10 mL volume bottle. After vortex oscillation, the solution was filtered through a 0.45 μm microporous filter membrane. A standard solution was prepared and mixed to form a reserve solution, which was then serially diluted to different concentrations to measure the standard curve (Table 5).

Preparation of the test product solution: *Schisandra chinensis* from seven harvesting periods was dried to a constant weight and then ground. Subsequently, 0.125 g of the ground *Schisandra chinensis* was placed in a 25 mL test tube, to which 18 mL of methanol was added. The mixture was subjected to a water bath at 65 °C for 20 min, followed by ultrasonic extraction (300 W, 40 kHz) for another 20 min. Finally, the extract was obtained through a 0.45 μm micropore filtration membrane.

The chromatographic conditions were as follows: Agilent ZORBAX C18 column (Agilent Technologies Co., Ltd., Santa Clara, CA, USA) (4.6 mm × 250 mm, 5 μm) with mobile phase water (A), methanol (B) at the flow rate of 1 mL/min, sample size of 10 μL, use UV detector, the UV detection wavelength of 220 nm, column temperature of 35 °C. Gradient elution: 0–20 min, A 45%, B 55%; 20.1–40 min, A 25%, B 75%; 40.1–47 min, A 22%, B 78%; 47.1–52 min, A 5%, B 95%; 52.1–60 min, A 45%, B 55%.

#### 3.3.2. Detection of Organic Acid Content

Preparation of reference solution: reference citric acid 10.08 mg, quinic acid 9.80 mg, L-tartaric acid 9.79 mg and L-malic acid 15.81 mg were precisely weighed, and 0.1% phosphoric acid aqueous solution was added into 10 mL volumetric bottle. After vortex shaking, 0.45 μm microporous filter membrane was used to prepare reference solution and mixed to make reserve solution. It was diluted step by step into different concentrations of reference solution to measure the standard curve (Table 6).

Preparation of the test product solution: the *Schisandra chinensis* from 7 harvesting periods was dried to constant weight, then crushed and ground, 0.5 g *Schisandra chinensis* was extracted in 25 mL test tube, 20 mL methanol was added, ultrasonic extraction (300 W, 40 kHz) for 40 min, and 0.45 μm micropore filtration membrane was used.

Chromatographic conditions: Eternal XT-C18 column (4.6 mm × 250 mm, 5 μm) mobile phase 0.1% phosphoric acid-water-methanol, flow rate 0.4 mL/min, sample size 10 μL, use UV detector, the UV detection wavelength 210 nm, column temperature 25 °C [22].

#### 3.3.3. Detection of Volatile Flavor Substances

Sample pretreatment: 0.5 g of *Schisandra chinensis* from 7 stages was placed in 20 mL headspace bottle, and 4-methyl-2-amyl alcohol was added into the sample as the internal standard substance.

HS condition: automatic headspace injection was used, the injection volume was 500 μL, the incubation time was 15 min, the incubation temperature was 60 °C, the injection needle temperature was 85 °C, and the incubation speed was 500 rpm.

GC conditions: MXT-WAX column (15 m × 0.53 mm, 1 μm), analysis time 46 min, column temperature 60 °C, carrier gas/drift gas was high purity nitrogen (99.999%).

IMS conditions: the drift tube length was 98 mm and the IMS temperature was 45 °C.

The concentration of 4-methyl-2-amyl alcohol was 198 ppb, the volume of signal peak was 540.78, and the intensity of each signal peak was about 0.366 ppb. The quantitative calculation of volatile flavor substances is as follows:$$Ci=\frac{Cis\times Ai}{Ais}$$

In the formula, Ci represents the mass concentration of any component—μg/Kg; Cis is the mass concentration of the internal standard used—μg/Kg; Ai/Ais is the volume ratio of any signal peak to the internal standard signal peak. For qualitative analysis of substances, the built-in NIST database and the IMS database were utilized.

#### 3.3.4. OAV Value Calculation

The OAV values of volatile flavor substances in *Schisandra chinensis* fruit were analyzed. It is generally believed that flavor substances with OAV value greater than 1 contribute more to fruit aroma and play a key role in the aroma system. OAV values are calculated as follows:$$OAV=\frac{Cx}{OTx}$$

In the formula, Cx represents the concentration of the volatile flavor substance x—μg/Kg; OTx is the threshold value of the volatile flavor substance in the air, mainly with reference to the *Odor Threshold: Compilation of Odor Thresholds in Air, Water, and Other Media* (2011 edition).

#### 3.3.5. Data Processing and Analysis

In the experiment, Excel 2010 was utilized for the statistical collation of the obtained data. SPSS 22.0 was employed for analyzing data differences. The Simca 14.1 software was used for OPLS-DA and VIP value analysis, while the Reporter plug-in was applied to compare the spectral differences between samples. Additionally, the Gallery Plot plug-in was engaged to examine the fingerprints. The differences in volatile flavor substances between samples were assessed, and the results were graphically represented using Origin 2021. We analyze the obtained data using OPLS-DA, OAV values, and VIP values [18].

## 4. Conclusions

Through the determination of lignan content in *Schisandra chinensis* fruit, it was found that the content of schisandrin in fruit from different harvesting periods had reached the pharmacopoeia standard. Throughout the entire growth and development period of the fruit, schisandrin, schisandrol B, schisantherin A, schisantherin B, schisandrin A, and schisandrin B exhibited a pattern of initially decreasing and then increasing, with the lignan content peaking in July. As the seed expansion ceased and the flesh cells began to expand, the lignan content fell to its lowest at the end of August, only to rise again in September. For the purpose of utilizing lignans as raw materials, it was advisable to select *Schisandra chinensis* fruit with a green skin in late July for harvesting, as the lignan content was higher at this time.

The organic acid content of *Schisandra chinensis* fruit was determined, and the results indicated that the organic acid content exhibited an increasing trend as the fruit matured, culminating in the highest level on 6 September. The correlation analysis revealed that the predominant organic acids in *Schisandra chinensis* fruit were primarily citric acid, whose content steadily increased throughout the entire growth period. Therefore, if organic acid is to be used as the main raw material for development, harvesting should be optimally carried out in early September, when the organic acid content in *Schisandra chinensis* fruit is at its peak.

In the study, HS-GC-IMS technology was utilized to compare and analyze the volatile flavor substances of *Schisandra chinensis* fruit at different harvesting stages, and the fingerprint of volatile flavor substances at these stages was established. A total of 67 volatile flavor substances were identified, with terpenes being the main category, and terpinolene was selected as the key volatile flavor substance influencing the aroma characteristics of *Schisandra chinensis* fruit at various picking stages by combining OAV value, VIP value, and *p*-value. Should the aroma component content of *Schisandra chinensis* fruit be considered for development and utilization, harvesting should optimally be carried out in early August, when the aroma component content is found to be higher.

## Figures and Tables

**Figure 1 molecules-29-01893-f001:**
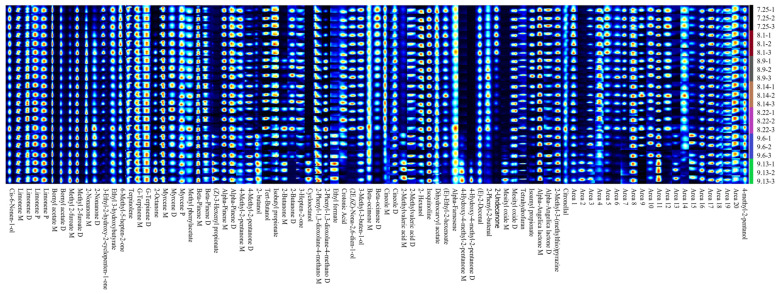
Fingerprint of volatile flavor substances in fruit of *Schisandra chinensis* in different harvesting periods.

**Figure 2 molecules-29-01893-f002:**
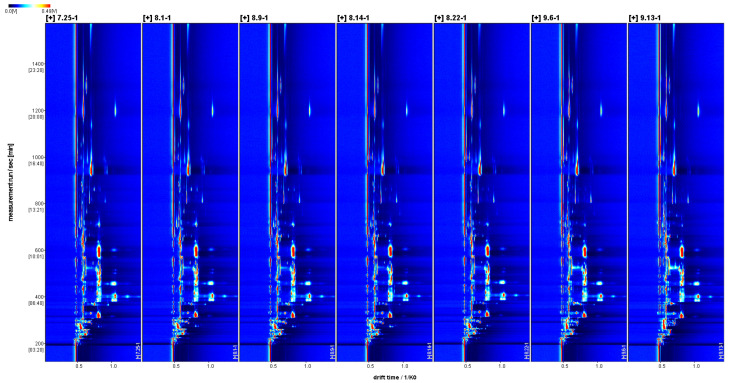
Two-dimensional spectra of volatile flavor substances in fruit of *Schisandra chinensis* at different harvesting stages (top view).

**Figure 3 molecules-29-01893-f003:**
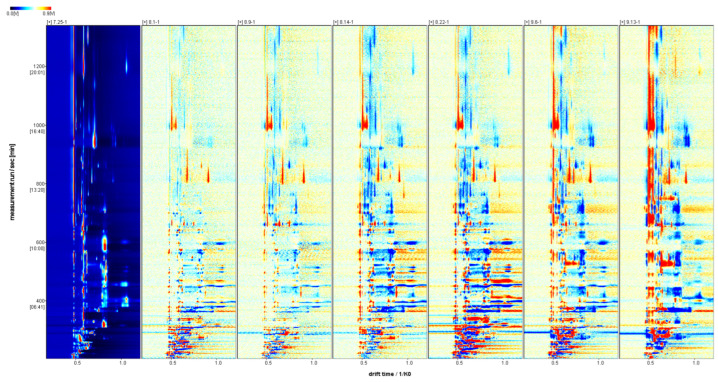
The difference spectra of volatile flavor substances in fructus *Schisandra chinensis* in different harvesting periods.

**Figure 4 molecules-29-01893-f004:**
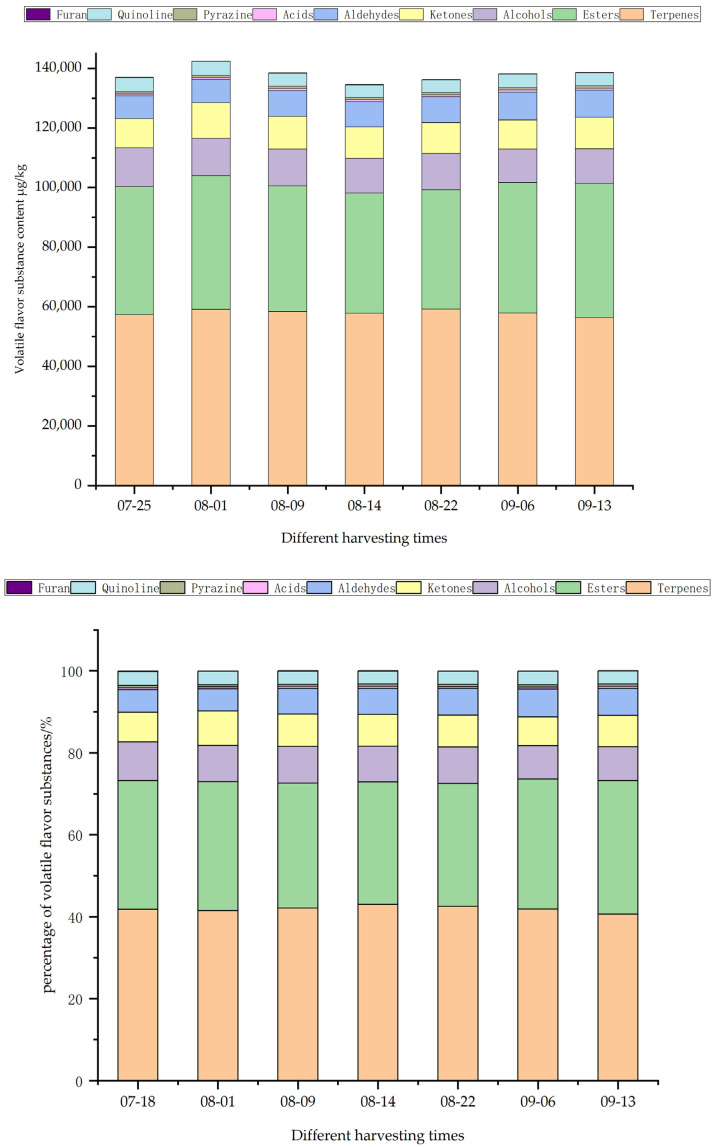
Content and percentage of volatile flavor substances in *Schisandra chinensis* in different harvesting periods.

**Figure 5 molecules-29-01893-f005:**
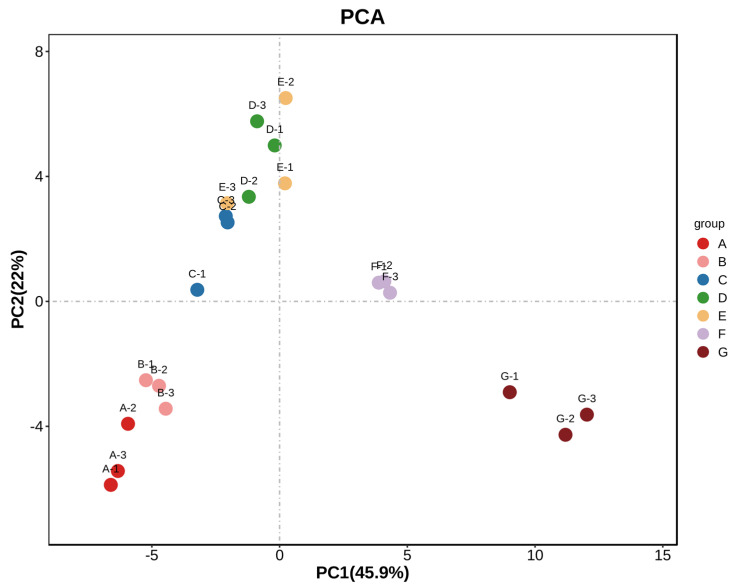
PCA analysis of volatile flavor compounds in fruit of *Schisandra chinensis* at different harvesting stages.

**Figure 6 molecules-29-01893-f006:**
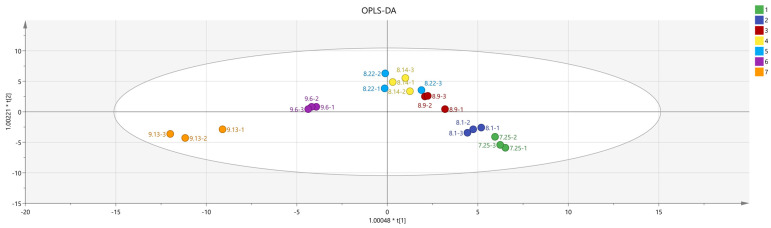
QPLS−DA analysis of volatile flavor compounds in fruit of *Schisandra chinensis* at different harvesting stages.

**Figure 7 molecules-29-01893-f007:**
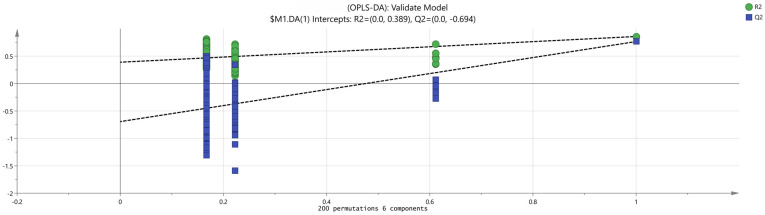
Results of cross−examination of volatile flavor substance models in different harvesting periods of *Schisandra chinensis*.

**Figure 8 molecules-29-01893-f008:**
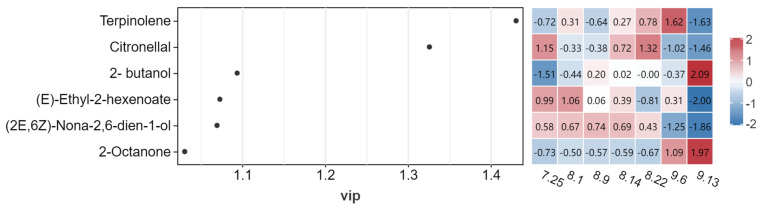
Analysis of VIP value of volatile flavor substances in fruit of *Schisandra chinensis*.

**Figure 9 molecules-29-01893-f009:**
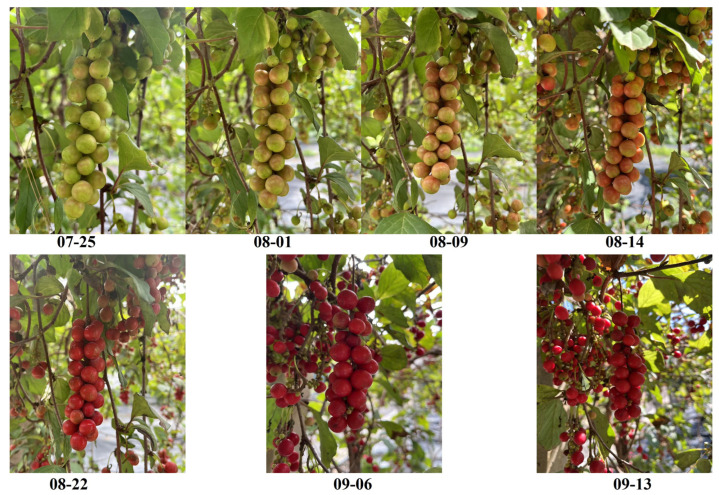
Fruit of *Schisandra chinensis* at harvest time.

**Table 1 molecules-29-01893-t001:** Lignans in fruit of *Schisandra chinensis* at different harvesting periods.

Date	Schisandrin (mg/g)	Schisandrol B (mg/g)	Schisantherin A (mg/g)	Schisantherin B (mg/g)	Schisandrin A (mg/g)	Schisandrin B (mg/g)	Total Lignans (mg/g)
7.25	21.70 ± 0.13 a	1.34 ± 0.01 a	3.92 ± 0.04 a	0.89 ± 0.02 a	2.34 ± 0.01 a	4.79 ± 0.03 a	35.00 ± 0.11 a
8.1	19.13 ± 0.11 b	1.18 ± 0.03 b	3.32 ± 0.08 b	0.75 ± 0.03 b	1.81 ± 0.02 b	4.22 ± 0.06 b	30.41 ± 0.31 b
8.9	17.34 ± 0.11 c	1.05 ± 0.01 c	2.89 ± 0.06 c	0.67 ± 0.02 c	1.55 ± 0.02 c	3.49 ± 0.07 c	27.00 ± 0.29 c
8.14	13.17 ± 0.07 d	0.81 ± 0.01 d	2.10 ± 0.01 d	0.48 ± 0.00 d	1.09 ± 0.01 d	2.93 ± 0.06 d	16.96 ± 0.18 d
8.22	10.88 ± 0.08 e	0.71 ± 0.02 e	1.75 ± 0.02 e	0.43 ± 0.01 e	0.98 ± 0.03 e	2.22 ± 0.04 e	14.11 ± 0.06 e
9.6	10.25 ± 0.09 f	0.64 ± 0.01 f	1.77 ± 0.00 e	0.41 ± 0.00 e	1.01 ± 0.01 e	2.24 ± 0.01 e	17.06 ± 0.09 f
9.13	10.71 ± 0.02 e	0.71 ± 0.03 e	1.83 ± 0.04 e	0.49 ± 0.04 d	1.01 ± 0.01 e	2.30 ± 0.02 e	16.96 ± 0.18 e

Note: Means with different letters in the same column express significant differences (Duncan’s test *p* < 0.05).

**Table 2 molecules-29-01893-t002:** Organic acids in fruit of *Schisandra chinensis* at different harvesting stages.

Date	Quinic Acid (mg/g)	L-Tartaric Acid (mg/g)	L-Malic Acid (mg/g)	Citric Acid (mg/g)	Total Organic Acid (mg/g)
7.25	0.33 ± 0.01 d	1.32 ± 0.03 c	29.20 ± 0.26 c	16.92 ± 0.13 f	47.77 ± 0.41 f
8.1	0.52 ± 0.04 a	1.27 ± 0.08 c	26.96 ± 0.19 d	19.54 ± 0.17 e	48.30 ± 0.38 f
8.9	0.46 ± 0.01 bc	1.33 ± 0.10 c	26.50 ± 1.45 d	22.23 ± 0.85 d	50.53 ± 2.33 e
8.14	0.45 ± 0.01 bc	1.36 ± 0.05 c	30.66 ± 0.50 b	27.45 ± 0.11 c	59.92 ± 0.96 c
8.22	0.42 ± 0.03 c	1.29 ± 0.02 c	26.94 ± 0.17 d	27.73 ± 0.05 c	56.39 ± 0.22 d
9.6	0.48 ± 0.01 b	1.73 ± 0.12 a	36.20 ± 0.52 a	33.93 ± 0.38 a	72.34 ± 0.61 a
9.13	0.44 ± 0.03 bc	1.53 ± 0.07 b	30.72 ± 0.21 b	30.59 ± 0.66 b	63.28 ± 0.81 b

Note: Means with different letters in the same column express significant differences (Duncan’s test *p* < 0.05).

**Table 3 molecules-29-01893-t003:** Correlation analysis of organic acids in fruit of *Schisandra chinensis* at different harvesting stages.

Organic Acid	Quinic Acid	L-Tartaric Acid	L-Malic Acid	Citric Acid
Quinic acid	1			
L-tartaric acid	0.25	1		
L-malic acid	−0.01	0.806 **	1	
Citric acid	0.531 **	0.705 **	0.475 *	1

Note: * indicates significant correlation at the 0.05 level; ** indicates highly significant correlation at the 0.01 level.

**Table 4 molecules-29-01893-t004:** OAV analysis of main aroma substances of *Schisandra chinensis* fruit at different harvesting stages (μg/kg).

NO.	Aromatic Substances	Class of Chemical Substance	7.25	8.1	8.9	8.14	8.22	9.6	9.13
	Terpenes								
1	alpha-Pinene	Terpenes	34.69	34.31	32.83	31.39	32.87	30.73	28.28
2	Terpinolene	Terpenes	11.01	11.23	11.03	11.22	11.33	11.50	10.82
3	Myrcene	Terpenes	20.61	22.04	22.34	23.14	23.09	22.84	20.52
4	G-Terpinene	Terpenes	1.62	1.66	1.71	1.74	1.74	1.74	1.75
5	Limonene	Terpenes	69.58	71.30	72.83	74.91	74.04	76.01	75.14
	Esters								
1	(E)-Ethyl-2-hexenoate	Esters	1.03	1.04	0.81	0.89	0.62	0.87	0.34
2	Bornyl acetate	Esters	14.40	15.31	13.89	12.82	13.63	14.01	13.96
3	Isobutyl propanoate	Esters	6.78	6.78	5.18	5.52	4.22	4.71	2.83
	Alcohols								
1	(2E,6Z)-Nona-2,6-dien-1-ol	Alcohols	2544.53	2625.04	2682.15	2644.71	2419.51	990.71	468.78
2	Cineole	Alcohols	712.27	700.22	728.09	701.71	729.44	570.45	495.72
3	2- butanol	Alcohols	0.18	0.27	0.25	0.24	0.21	0.87	1.21
	Ketone								
1	6-Methyl-5-hepten-2-one	Ketone	45.81	46.08	33.55	32.26	29.17	24.53	20.48
2	2-Nonanone	Ketone	25.22	43.13	43.17	39.57	47.40	46.02	34.11
3	Mesityl oxide	Ketone	2.25	2.23	2.46	2.56	2.73	2.47	2.66
4	2-Octanone	Ketone	0.32	1.13	1.61	1.48	1.46	1.18	3.05
	Aldehyde								
1	Citronellal	Aldehyde	499.39	425.55	422.80	478.13	507.70	390.89	369.23

**Table 5 molecules-29-01893-t005:** Lignan standard curve.

Standard Name	Standard Curve	Correlation Coefficient
Schisandrin	Y = 12,767x − 95.773	R^2^ = 0.9997
Schisandrol B	Y = 4721.9x − 30.978	R^2^ = 0.9999
Schisantherin A	Y = 3428.7x − 37.981	R^2^ = 0.9998
Schisantherin B	Y = 3760.7x − 30.306	R^2^ = 0.9999
Schisandrin A	Y = 5883.3x − 49.786	R^2^ = 0.9998
Schisandrin B	Y = 3725.3x − 29.445	R^2^ = 0.9999

**Table 6 molecules-29-01893-t006:** Organic acid standard curve.

Standard Name	Standard Curve	Correlation Coefficient
Quinic acid	Y = 989.63x − 43.327	R^2^ = 0.9998
L-tartaric acid	Y = 3526.5x − 23.307	R^2^ = 0.9999
L-malic acid	Y = 2396.1x + 6.6547	R^2^ = 0.9998
Citric acid	Y = 844.37x − 49.077	R^2^ = 0.9999

## Data Availability

Data are available within the manuscript and Appendix A.

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
