# Peer review of "Based on HPLC and HS-GC-IMS Techniques, the Changes in the Internal Chemical Components of Schisandra chinensis (Turcz.) Baill. Fruit at Different Harvesting Periods Were Analyzed"

_molecules, 2024, doi:10.3390/molecules29081893_

Round 1

Reviewer 1 Report

Comments and Suggestions for Authors

The manuscript presented by the authors meets the journal's scope and possesses the scientific rigor to be considered for publication in Molecules. However, substantial corrections are required before it is approved for publication. Below are my comments:

The manuscript title must include the complete name of the species: Schisandra chinensis (Turcz.) Baill.

In the abstract, the authors mention that the species Schisandra chinensis is part of Chinese medicinal herbs and has various medicinal uses. What is the primary medicinal use of this species? This information should be highlighted.

Does “pharmacologically active components” in the abstract refer to all groups of phytochemicals? Seasonal variation may alter some secondary metabolites, but I believe the focus should be narrowed to one group, as this section is unclear.

It is mentioned that HPLC was used, but the detector is omitted. Please add the complete technique.

It should be clarified which chemical species or group of phytochemicals is referred to when mentioning “volatile flavor substances” since using Headspace limits this group of phytochemicals to highly volatile ones. Introducing the vial homogenization temperature range and the syringe temperature would be worthwhile. This can be addressed in this comment and highlighted in your methodology.

The authors do not present a conclusion of their study in the abstract, only referencing 16 flavor substances and identifying Terpinolene as the essential volatile flavor. There is no mention of the phenological cycle or seasonal variation. Nor is it mentioned if the Schisandra chinensis trees were of the same age and the vegetative stage they were in. It is imperative to clarify this.

Please ensure that Schisandra chinensis is written correctly throughout the introduction, respecting uppercase and lowercase, and is italicized.

Review the journal's template; references should be in brackets [] in the text, not as superscripts.

A quick search on PubMed with the word “Schisandrin A” yields exciting results, including studies on obesity, anti-inflammatory, renal fibrosis, cancer, and pharmacokinetic studies of the molecule. Consider elaborating more on the state of the art of the properties of this compound.

In the results section, the authors state, “Lignan is the main pharmacological active ingredient in Schisandra chinensis,” first, please ensure that Schisandra chinensis is correctly written. Second, Lignan is a group of polyphenols; could you specify which ones are the predominant ones?

The way the results are presented is correct; however, I believe the tables complicate visual comparisons, so I suggest switching the tables for bar graphs. This will help provide a quicker overview of the sampling dates and metabolic content and understand the significant differences. I leave this to the authors' discretion but request that content not be repeated.

Figure 1 is attractive and helps understand the metabolome through the heat map; however, in such figures, authors should aim to separate the figure by similar analytes or functional groups and increase the resolution, as it is not clear. This also applies to Figures 2 and 3.

The discussion and conclusions are relevant to the results.

The authors have presented interesting work on the model Schisandra chinensis (Turcz) Baill. The proposal meets the scope of the journal and is well-organized with a clear and robust methodology. I ask the authors to address my comments to approve the manuscript. For now, I will give major mandatory recommendations, intending to review the manuscript again once the requested changes have been made.

Reviewer 2 Report

Comments and Suggestions for Authors

Comments to the Author
Dear Editors and Authors,
Regarding the
Manuscript ID: molecules-2958443, entitled : Based on HPLC and HS-GC-IMS techniques, the changes of internal chemical components of schisandra chinensi fruit at different harvesting periods were analyzed “  my comments are the following:

The research on the effects of
harvesting periods on chemical components of schisandra chinensi fruit is very interesting. The introduction provides sufficient background regarding the research subject of the research. Discussion of results provides valuable information in order to draw a safe conclusion about the studied material.

1.      Line 30: please added semicolon after lignans

2.      Line 48: Please added references

3.      Line 98: Please added references

4.      Line 333: 7.25, 8.1, 8.9, 8.14, 8.22, 9.6, and 9.13; Clarify the mentioned numbers

5.      Line 336: The indices are measured: What do you mean?

6.      Experimental Methods: Added references, the analyses were carried out using what methods?

7.      Tables  do not have captions, (legends, units…) and Figures must be presented in such way to be more readable

8.      In the methodology section,  can you added the number of samples used and the year of experimentation

9.      The significance of the correlation in Table 5 (Pearson correlation coefficients) should be indicated.

10.  It is necessary to revise English thought the manuscript: The manuscript is full of naive mistakes.

11.  Why did you use specially the Yan zhihong Variety? What is the most abundant variety in China?

12.  Why did you limit your analyses to lignan? what do you think about the other phenolic classes?

Based on the above, I suggest the acceptance of the manuscript with minor revision.

Comments on the Quality of English Language

It is necessary to revise English thought the manuscript: The manuscript is full of naive mistakes.

Round 2

Reviewer 1 Report

Comments and Suggestions for Authors

The authors have addressed each of my comments and provided a correct response. The manuscript's title and introduction have been improved, clearly stating the problem and the importance of their work. The authors correctly modified their tables and provided bar graphs to understand the phytochemical composition and its variation over time. I don't have any more comments to add to the manuscript. I appreciate the authors attending to my observations and recommend that the manuscript be accepted for publication.